# Optimising reporting of adverse events following immunisation by healthcare workers in Ghana: A qualitative study in four regions

**Raymond Akawire Aborigo**[1]*, **Paul Welaga**[1], **Abraham Oduro**[1], **Anna Shaum**[2], **Joseph Opare**[3], **Alex Dodoo**[4], **Hilda Ampadu**[4], **Jane F. Gidudu**[2]

**1** Navrongo Health Research Centre, Research and Development Division, Ghana Health Service, Navrongo, Ghana, **2** Center for Global Health, Global Immunisation Division, Centers for Disease Control and Prevention, Atlanta, GA, United States of America, **3** The Ghana Second Year of Life Vaccine Platform (2YL), Accra, Ghana, **4** The African Collaborating Centre for Pharmacovigilance, Accra, Ghana

* rayborigo@yahoo.com

## Abstract

### Introduction

Despite the emphasis on reporting of Adverse Events Following Immunisation (AEFIs) during didactic training sessions, especially prior to new vaccine introductions, it remains low in Ghana. We explored the factors underlying the under-reporting of AEFI by healthcare workers (HCWs) to provide guidance on appropriate interventions to increase reporting.

### Methods

We conducted an exploratory descriptive in-depth study of the factors contributing to low reporting of AEFI among HCWs in four regions in Ghana. Key informant interviews (KII) were held with purposively selected individuals that are relevant to the AEFI reporting process at the district, regional, and national levels. We used KII guides to conduct in-depth interviews and used NVivo 10 qualitative software to analyse the data. Themes on factors influencing AEFI reporting were derived inductively from the data, and illustrative quotes from respondents were used to support the narratives.

### Results

We conducted 116 KIIs with the health managers, regulators and frontline HCWs and found that lack of information on reportable AEFIs and reporting structures, misunderstanding of reportable AEFIs, heavy workload, cost of reporting AEFIs, fear of blame by supervisors, lack of motivation, and inadequate feedback as factors responsible for underreporting of AEFIs. Respondents suggested that capacity building for frontline HCWs, effective supervision, the provision of motivation and feedback, simplification of reporting procedures, incentives for integrating AEFI reporting into routine monitoring and reporting, standardization of reporting procedures across regions, and developing appropriate interventions to address the fear of personal consequences would help improve AEFI reporting.

**Data Availability Statement:** All relevant data are within the manuscript and the Supporting Information files (WINZIPPED). The complete dataset is stored in an institutional server and can

be accessed through the institutional contact: Peter.wontuo@navrongo-hrc.org.

**Funding:** • Initials of the authors who received each award: RAA • Grant numbers awarded to each author - Subcontract No: AF-NRC001/2019 • The full name of each funder: Centers for Disease Control and Prevention/the Agency for Toxic Substances and Disease, Atlanta, USA, Through the African Field Epidemiology Network (AFINET) • URL of each funder website: https://www.cdc.gov • Did the sponsors or funders play any role in the study design, data collection and analysis, decision to publish, or preparation of the manuscript? YES - Specify the role(s) played.: They participated in study design and in the decision to publish.

**Competing interests:** The authors have declared that no competing interests exist.

## Conclusion

From the perspectives of a broad range of key informants at all levels of the vaccine safety system, we found multiple factors (both structural and behavioural), that may impact HCW reporting of AEFI in Ghana. Improvements in line with the suggestions are necessary for increased AEFI reporting in Ghana.

## Introduction

Although immunisation has been shown to be one of the most successful and cost-effective ways to save lives by preventing numerous diseases, the delivery of vaccines requires strong health systems [1, 2]. Vaccine safety systems are a critical component of the larger health system because they allow authorities to address safety concerns promptly and promote confidence in immunisation programs, especially when new vaccines are introduced. Paying attention to the efficient functioning of these safety systems is critical as more vaccines are introduced into low- and middle-income countries (LMICs) [3, 4]. In most LMICs, the reporting of adverse events following immunisation (AEFI) is sub-optimal—below the minimum expected reporting ratio of 10 reported AEFIs per 100,000 surviving infants [5] which is a standard established by the World Health Organisation (WHO) for countries with minimal capacity for vaccine safety monitoring. All AEFIs encountered should be reported to the country surveillance system to determine the country reporting ratio. WHO's Global Vaccine Safety Initiative (GVSI) has recently focused on improving minimum capacity for vaccine safety in LMICs, particularly in Africa where AEFI surveillance is sub-optimal, and strengthening AEFI surveillance systems is in line with this initiative [6].

To protect children from vaccine-preventable diseases (VPD), Ghana is strengthening the second year of life (2YL) immunisation platform, using a multi-faceted approach with six priority areas identified by the stakeholders [7]. To meet unique challenges to accessing and utilizing routine immunisation services, this approach includes efforts to improve VPD surveillance, including monitoring AEFIs and developing innovative strategies [3].

A baseline survey of health facilities and households was conducted in the three regions (Northern, Greater Accra, and Volta) of Ghana in 2016 to identify and address barriers contributing to low uptake of the second dose of measles-containing vaccine (MCV). One of the findings from this survey identified the fear of AEFIs as an important factor in caregivers' avoidance of vaccinations [8]. Lack of adequate AEFI monitoring and appropriate response to caregivers' concerns may cause parental distrust in immunisation programs, potentially resulting in the lack of vaccine acceptance, reduced coverage, and VPD outbreaks [9].

In Ghana, a national electronic data system for AEFI reporting was initiated in 2002, following the introduction of the diphtheria-tetanus-pertussis-hepatitis B-*Haemophilus influenzae* type b combined vaccine (commonly called pentavalent vaccine). AEFIs are monitored jointly by the Ghana Food and Drug Authority (FDA) and the Expanded Programme on Immunisation (EPI) in close collaboration with the African Collaborating Centre for Pharmacovigilance (ACC). Recently, new methods for AEFI surveillance (e.g. using disproportionality analysis reporting ratios to identify safety signals) have been implemented [10]. FDA and EPI conducted didactic AEFI training courses for HCWs in Ghana for the last six years, especially when introducing new vaccines. However, the reporting of AEFI has remained low (reporting ratio of 1.65 per 100,000 surviving infants and only 13 AEFIs reported in 2015) [11]. Ghana is one of the three countries in Africa selected by WHO to pilot-test the new RTS,S malaria

vaccine before universal use, which reinforces the importance of addressing weaknesses in AEFI systems [12] by examining reasons for under-reporting of AEFIs in Ghana. Our study aimed to explore these factors with relevant individuals to provide guidance on appropriate interventions to improve AEFI reporting.

## Materials and methods

### Setting

Ghana is a tropical coastal country located in West Africa, bordered by Togo, Burkina Faso, and Cote d'Ivoire. It has a population of about 29 million with about 850,000 surviving infants each year (2019) and an annual infant mortality rate of 43/1000 (2015) [11]. Health services, including immunisation, are packaged and delivered in communities by clinics, health centers, and tertiary hospitals. Public facilities provide approximately 60% of the country's health service delivery [13]. Community-based health planning services (CHPS) are the lowest healthcare unit that provides immunisations.

### Study design

We conducted descriptive in-depth interviews with key informants to identify factors that contribute to the under-reporting of AEFI by frontline HCWs in four regions in Ghana. According to the 2010 population census, the four regions have a combined population of 9,654,312 [14]. Three of the regions—Northern, Volta, and Greater Accra—were selected because the 2014 Demographic and Health Survey (DHS) revealed that these regions had low coverage of second-dose MCV [15], and they were part of an ongoing project to improve vaccination coverage during 2YL [9]. Upper East Region was included in the study to complement WHO efforts to strengthen AEFI surveillance in areas where the RTSS malaria vaccine was introduced. As part of the larger study to identify factors associated with underreporting of AEFI, we conducted cross-sectional surveys on AEFI reporting among HCW across 20 districts concurrently; these results have been published separately [16].

### AEFI reporting structure in Ghana [17]

Surveillance of AEFIs in Ghana is a collaborative effort of the Expanded Program on Immunization, the Food and Drugs Authority (FDA), the World Health Organisation and UNICEF. Other stakeholders include the vaccine recipients, parents and caregivers, community members, civil society, private health providers, the media and the general public. The aim of the surveillance is to promptly detect and manage AEFIs whether real or perceived. According to the FDA guidelines, a suspicion alone is enough grounds for reporting [17].

From Fig 1, the flow of reporting varies depending on whether the AEFI emanated from a routine vaccination or a mass immunization campaign. Generally, four levels of reporting are used during mass immunization campaigns. The health worker reports to the disease control officer at the district level and the district officer reports to the disease control officer at the regional level and the regional officers report to the EPI at the national level. Routine vaccinations tend to follow the Ghana Health Service (GHS) structure which is from the health worker to the district disease control officers, to the regional EPI coordinators and then to the national EPI AEFI coordinator and finally to the FDA.

From the diagram, vaccine recipients and caregivers report all AEFIs to their health care providers. Health care providers or vaccinators are the lowest administrative level in the AEFI surveillance system. They communicate possible adverse events and how to manage mild and common reactions to vaccinees and /or caregivers before vaccination. They also detect,

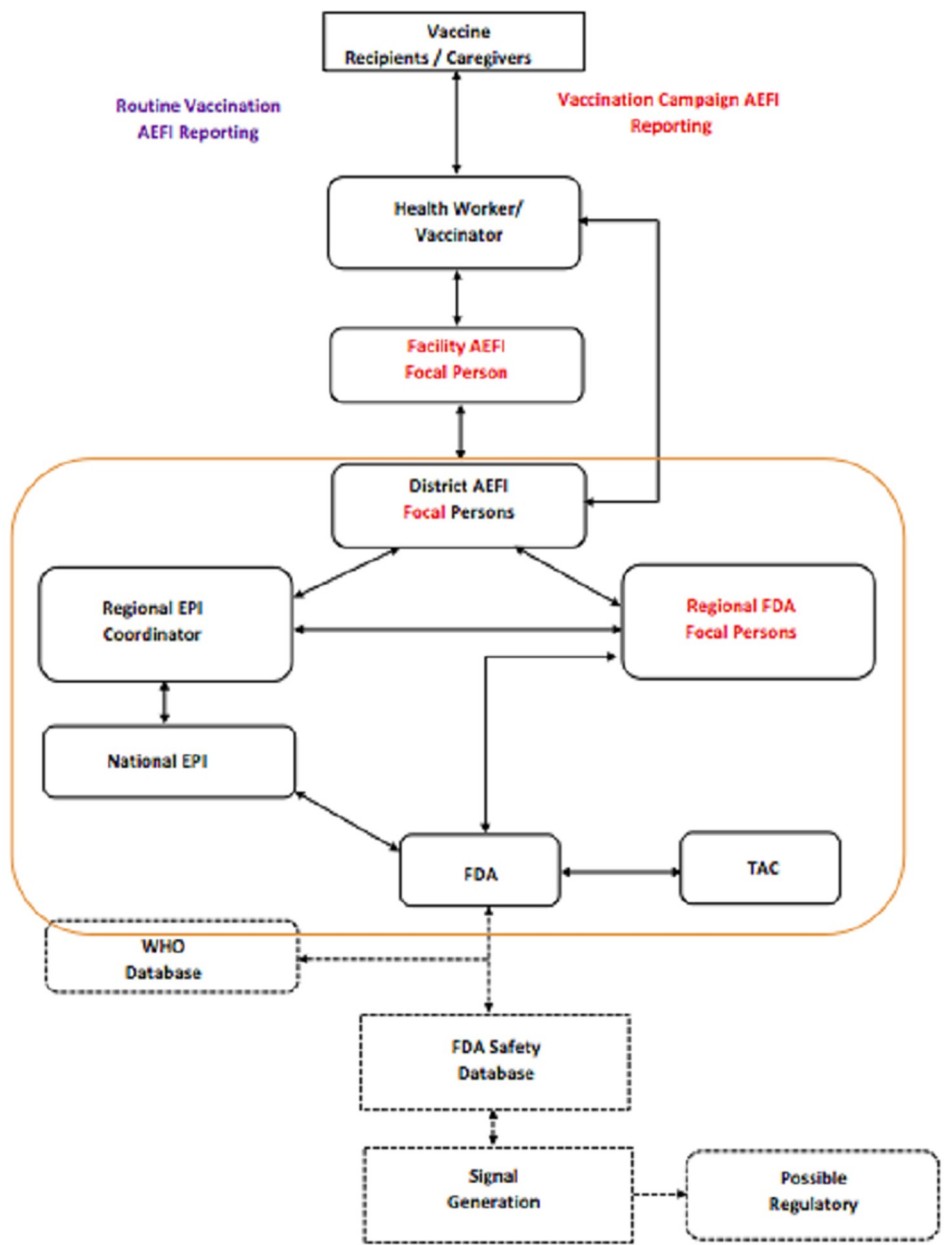

**Fig 1. AEFI reporting in Ghana (adapted from the Ghana FDA) [17].**

manage and report AEFIs using the guidelines. The Facility AEFI focal person is a trained health worker based in a clinic or hospital who sensitises all health workers to detect, manage and report AEFIs, conduct clinical investigations, report AEFIs and compile weekly AEFI reports for submission to the district AEFI Focal Person. The district AEFI Focal Person is a surveillance officer or health worker who is designated by the district health authorities as a focal person for AEFIs and has received training from the FDA and the EPI. The district AEFI Focal Person among other things, validates AEFI reports, maintains an AEFI database at the district level, perform analysis on AEFI data to determine distribution and patterns of occurrence, compile the reports from the community and health facilities and submit same to the

Regional EPI Coordinator and/or the FDA regional Focal person. The District AEFI Focal Person also submits daily AEFI line-listings to the FDA Regional Focal Person and relevant levels during vaccination campaigns. The Regional EPI Coordinator among other things maintains a regional database of AEFIs, analyses regional AEFI data to determine distribution and patterns of occurrence, compiles monthly AEFI reports from the districts and submits same to the national EPI. The Regional EPI Coordinator leads public communication on AEFIs at the regional level and assists the FDA Regional Focal Person to collate AEFI reports during mass vaccination campaigns. The FDA regional Focal Person among other things collects, validates and ensures the completion of reports from reference hospitals, gathers and qualifies reports from the District Focal Person, forwards all reports to the central FDA and ensures compliance with standard operating procedures for AEFI reporting during mass immunization campaigns. For routine surveillance, the Expanded Program on Immunizations /Ghana Health Service (EPI/GHS) designs, establishes, maintains and evaluates the surveillance system in conjunction with the FDA. They revise and update reporting tools and make them accessible. They also maintain a database at the national EPI office, analyse the data, provide feedback and support to the districts and regions. They submit AEFI reports from routine immunizations to the FDA, respond to rumours and manage crisis. They also provide data on vaccine performance on a regular basis to the FDA.

The FDA assists the EPI/GHS to develop reporting tools, constitutes an expert committee to evaluate the reports, analyse and provide feedback to all stakeholders. They assist in training personnel involved in AEFI reporting, share the AEFI information with international agencies and manufacturers and carry out risk benefit analysis of vaccines used in immunization programs. Within the reporting framework, only the FDA determines causality and feedback to all levels. The FDA constitutes the Technical Advisory Committee (TAC) which also receives reports for safety evaluation and causality assessment if necessary. The committee submits recommendations to the FDA and the EPI for consideration [17].

## Study population

We interviewed the frontline HCW, members of the district and regional health management teams (DHMT & RHMT), and FDA and EPI coordinators at the district, regional, and national levels.

## Sampling

This study was part of a larger study that sampled 176 health facilities across 20 participating districts for a quantitative AEFI assessment [16]. Using purposive sampling, we identified HCWs who reported ever encountering an AEFI from the quantitative sample. We conveniently selected one HCW from each of the 10 health facilities in each of the selected study districts for in-depth interviews. We then asked them to share their experiences with regards to reporting AEFIs to the FDA or EPI to gain insights into why AEFIs are not reported and the challenges in the reporting process.

In addition, through purposive sampling, we identified district, regional and national level managers who were associated with AEFI reporting. In each region, we randomly selected five districts and conducted KIIs with two members of the DHMT (the public health nurse and the health information officer) and the EPI coordinator who, in most cases, was the disease control officer. Likewise, we selected two RHMT members (the regional surveillance and disease control officers). Additionally, two FDA staff at the regional offices, as well as the national office and one national EPI coordinator participated in KIIs. In total, the targeted number of KIIs to be conducted was 116 (Fig 2).

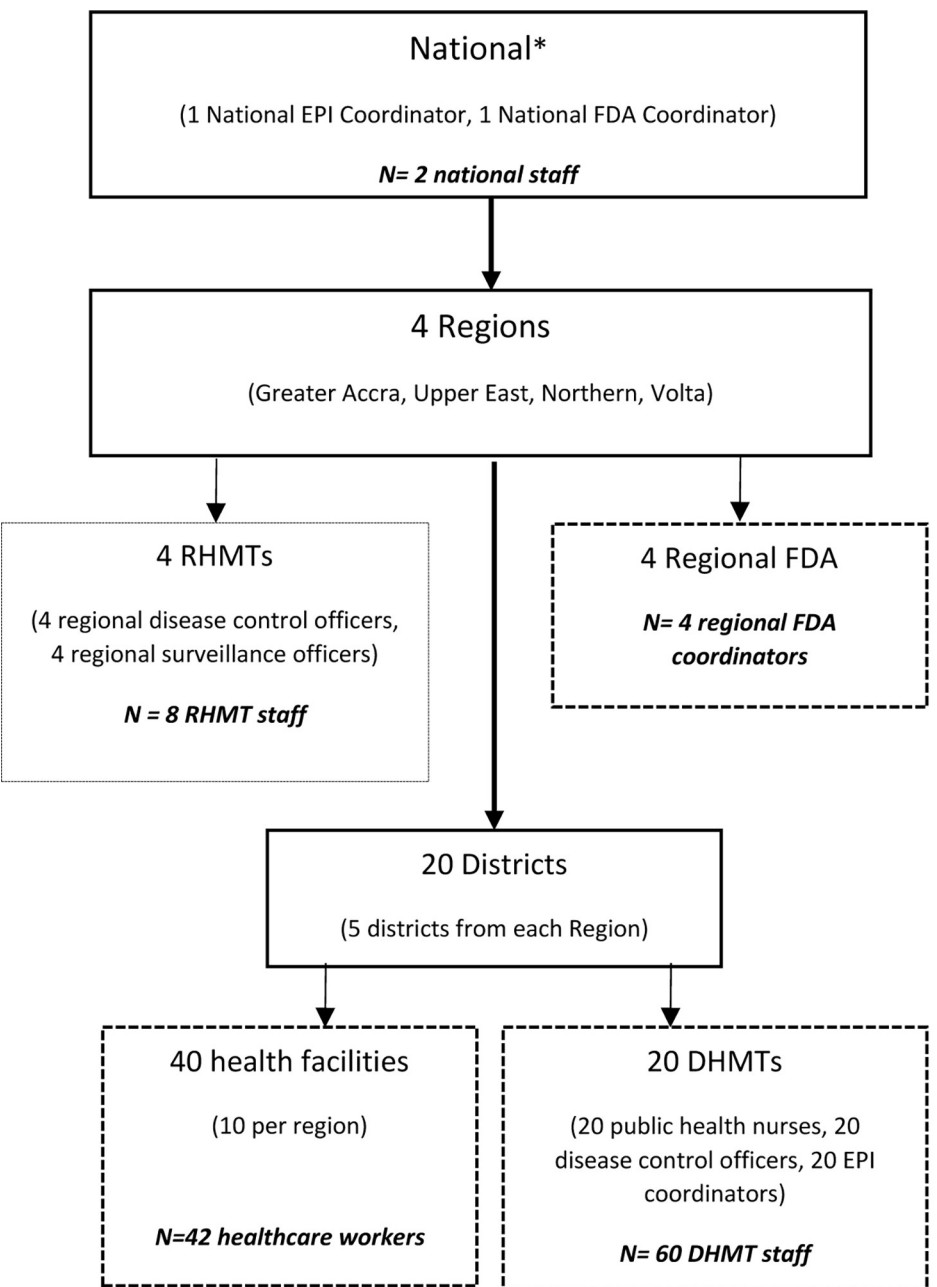

EPI = Expanded Programme on Immunization; FDA = Food and Drug Administration; RHMT = Regional Health Management Tema; DHMT = District Health Management Team

*Ghana's administrative structure flows from the national level to the regional level to the district level, to the health facility level.

**Fig 2. Approach to selecting respondents for key informant interviews.**

## Data collection

Collaborating with partners, we developed two interview guides based on the study objectives, pretested them and made the necessary revisions before use in the field. These two guides had subtle differences that reflected the role of the respondent, either as a manager or a frontline HCW.

**Table 1. Interviews conducted.**

| Target group | Number of interviews conducted |
|---|---|
| District Health Management Team (DHMT) | 40 |
| Regional Health Management Team (RHMT) | 08 |
| Food and Drug Administration (FDA) | 05 |
| Health care workers (HCW) | 42 |
| Expanded Programme on Immunization (EPI) | 21 |
| Total | 116 |

Fourteen graduate school-level data collectors with at least one-year of experience in qualitative research were trained for two-days to familiarize them with the purpose and methodology of the study. They honed their skills in seeking consent, the principles of qualitative interviewing, and producing a verbatim transcript. Data collectors conducted 116 KIIs from December 2017 to January 2018 with respondents at their work places in four regions.

Questions in the data collection tools asked respondents to describe their understanding of the AEFI surveillance system, issues related to the capacity of frontline HCWs to report AEFIs, barriers to reporting AEFIs, and suggestions for improvements. The interviews were conducted in English with each interview lasting between 30 to 50 minutes.

To determine thematic saturation and to identify new or emerging concepts, we compared the data collected each day with the previous data during daily debriefing sessions. When we reached the pre-determined number of interviews, the data across the regions became repetitive; therefore, there was no need for additional interviews.

The number of interviews that were conducted with each category of respondent is shown in Table 1.

## Data processing and analysis

All audiotapes were transcribed and processed, using Microsoft Word Office 365. RA, PW, and AO reviewed all the transcripts for accuracy and completeness before importing them into NVivo Qualitative Data Analysis Software; QSR International Pty Ltd. Version 12, 2018 for coding [18]. We performed the initial coding based on the deductive codes derived from the interview guide, segmenting the data into groupings to form preliminary categories of information on factors that contribute to the under-reporting of AEFIs. Subsequently, we derived themes inductively by reading all the transcripts and identifying normative patterns in the responses.

Using the deductive codes as a guide, two masters-level research officers and one of the authors (RA) conducted focused coding of the data. Discussions were held regularly among coders to agree on the meaning of ambiguous codes. A coding comparison query to determine the inter-rater reliability returned a Kappa co-efficient of 0.76, which represents a high level of agreement beyond chance.

## Ethics approval

This study was reviewed and approved by the CDC's Human Subject's Office and the Ghana Health Service's Ethical Review Committee. We explained the purpose of the study, the procedures involved, the risks and benefits, voluntariness of participation, the right to withdraw participation and the right to ask questions and receive reasonable answers to each potential respondent before the start of the interview. We received verbal consent from the respondents prior to their participation and sought consent to audio-record during the interviews.

## Findings

A total of 116 KII revealed that AEFIs are reported through three main ways: verbal, written, and electronic. Most facilities used paper-based forms to report, and the hierarchy for reporting AEFIs was similar across the regions as follows. The health facility reports to the EPI focal person for the district; then the district either reports directly to the EPI focal person at the regional level or the FDA officer for the region. Officers at the regional level report to the national office of the FDA; then FDA reviews all AEFI reports and provides feedback to the frontline HCW only when their assessment indicates that the adverse event is serious and related to the vaccine. We identified 13 themes in total; 6 were key factors contributing to underreporting and 7 were suggestions to improve reporting.

## Key factors contributing to under-reporting

**Frontline HCW's lack of capacity to report AEFIs.**   Two major challenges in AEFI reporting were identified among HCW responses: 1) recognizing the adverse events, and 2) knowing the reporting requirements and processes. The managers at all levels acknowledged that this had implications for the safety of their clients and the trust of the communities they serve. A health manager said:

"...*this can be affected by a lot of things such as some staff do not actually know anything about AEFIs; probably they have heard of it but they do not know when to report it and even how to report it. ...some have not even seen the form, how it looks like and even how to report on it."* **(DHMT-Northern Region)**

Several managers stated that the high attrition rate among trained staff and the lack of orientation for newly-posted HCW on AEFIs is another major limitation to AEFI reporting.

"...*maybe they were not given an orientation in that area, so it might be due to the training and the high attrition rate of staff. Now the new thing is, everybody is going to school, so after three years you go to school so the experienced ones are no more in the field doing the work. Most often they are new and you always have to start from scratch taking them through the processes."* **(DHMT-Volta Region)**

Educating and orienting the frontline HCW on AEFI reporting were recognised as the keys to effective monitoring and reporting of AEFIs. Respondents from the FDA reported that a series of trainings, targeting frontline HCWs nationwide have been held. However, the HCWs stated that the FDA trainings are usually held in the regional capitals, which limits participation to only health managers and heads of the facilities who do not share their knowledge with the frontline HCWs upon their return to post.

**Misunderstanding of reportable AEFIs.**   We found that respondents categorised AEFIs into serious and unexpected, and normal and expected. The majority of HCWs considered the former reportable, while the latter is not. AEFIs were considered serious if they were life-threatening, clinically significant, or severe and relevant to the patient's future vaccination. Those such as rashes, fever, and headaches were considered normal, easy to handle; hence, not reportable.

"... *it looks like we are not seeing very severe adverse events. They are minor ones, so our staff think that it is not necessary to report the minor ones. They give their paracetamol or whatever that will let it go and assure the people so they do not think that it is necessary to report;*

*even to fill a form and then alert us. . . .I think that is the main reason why we have low reporting.*" (**DHMT-Volta Region**)

**Workload.** The frontline HCWs indicated that they are already saddled with a lot of documentation in their daily practice; therefore, reporting of AEFIs adds to the burden. They complained about the complexity of the EPI AEFI reporting form, the volume of information required, and the amount of time needed.

"*. . . when the form is loaded, it puts the users away. . . .I think we need the information but to somebody at the lower level, they think taking time to fill all these forms is discouraging.*" (**HCW-Volta Region**)

Apart from the perceived burden associated with AEFI reporting, the frontline HCWs did not consider AEFI reporting as their core responsibility because they are not required to report these during official meetings. Furthermore, their supervisors do not request reports during monitoring visits; as a result, they prioritise activities that require periodic reports over AEFI reporting.

**Cost of reporting AEFIs.** Some HCWs identified the cost of tracing clients who report AEFIs, the cost of call credit to contact the next level of care, and delivering the form to their managers as barriers to reporting AEFIs. They also identified the unavailability of AEFI forms as another limiting factor for reporting.

"*Some years back, there used to be something we called imprest where they give the CHPS centre thirty Ghana cedis (6 USD). . . We use part of that money to do photocopies; you can see some of the louvers are broken and all that but for the past 5 years now, that is not forthcoming and at the end of the month you just have to pull out your money and go and do the photocopies. . .Sometimes the month ends and you don't have up to ten cedis with you; you know the reports are even bulky and so if you don't have up to ten cedis, you cannot write and those are the challenges.*" (**HCW-Upper East**)

**Fear of blame.** One overarching factor that was commonly reported was fear of being labeled incompetent, ignorant, non-performing, negligent, or even being scolded for not following protocols if the AEFI is linked to vaccine administration. Frontline HCWs said that they do not report because they often feel guilty, and whenever mothers return with an abscess or other life-threatening condition, this raises questions about their competence.

"*When we report on AEFIs, it looks like we are not doing well. It's like I have not done the right thing that is why these things are occurring. So definitely when I get or even when they bring me an AEFI, the tendency to report it is usually low because I think that if I report it, I will end up being scolded.*" (**HCW-Northern Region**)

A member of the FDA said that the HCWs think reports on AEFIs affect their performance appraisals negatively so they are fearful of completing them.

"*The other reason is that, still, people have not gotten to the level that your reporting does not actually lead to incrimination where you are liable to punishment or something and that when you do it, it is supposed to help the system. It is supposed to help re-design training and*

*education materials. People have not gotten to those terms yet because it is believed that when you do it either you will be castigated, seen as a conduit for creating problems for the district or some sort of hierarchy."* **(FDA-Volta Region)**

**Lack of motivation to report AEFIs.** HCWs reported that they are not motivated to report AEFIs because they do not know how the information will be used. They also expressed interest in getting feedback from the health managers and the FDA.

"*I have realized that feedback is very important. If I report to you and nothing comes back, then I am not motivated to report more. I am not even sure whether the report I have given to you is important to you*". **(HCW-Upper East)**

While the HCWs defined motivation as receiving feedback on the AEFI reports they submit, the health managers interpreted it as financial incentives.

"*I would say over the years there have been series of trainings given to a lot of health workers and we expect that they should be able to report all adverse events following Immunisation or dispensation of drugs in their duty. However, their reporting is so low or appalling so we want to attribute it to motivation. . . They do not take it as their core duty or mandate to report these adverse events to us even though we have institutional contact persons who were trained to report all adverse events. They have the forms to complete and forward them to us but we think that they are not being motivated enough. . . .So the under-reporting is just because they are not motivated. There should be a package put in place to motivate the health professionals to report AEFIs.*" **(FDA-Upper East)**

## Suggestions to improve AEFI reporting

**Capacity building and refresher trainings for frontline HCWs.** Respondents repeatedly suggested capacity building and refresher trainings for frontline HCWs to ensure efficient reporting of AEFIs. They emphasized that it is necessary to invite frontline HCWs to trainings and that their annual performance appraisals should be used to determine their training needs. They advised that community health volunteers should also be included in these trainings because they complement the HCW's efforts by actively surveying the community members for AEFIs and reporting them to the health facility heads.

"*My suggestion is continuous training of all the health providers on adverse events following Immunisation from the district level down to the CHPS compound level. Also, refresher training to some of the volunteers who are supporting us because some of these things are picked up by them when they come across them; they call the health staff to follow up. So I think continuous refresher training of these events is very key.*" **(DHMT-Upper East)**

All participants supported strategies for updating the knowledge of frontline HCWs via regular refresher trainings, workshops, and on-the-job coaching on AEFI reporting. They expressed the need for focused practical training on AEFI monitoring and reporting during orientations for newly-posted nurses.

**Effective supportive supervision.** Respondents suggested regular supportive supervision of the frontline HCWs to motivate them to report AEFIs by completing all the relevant forms.

There were suggestions that the district focal persons on AEFIs could make regular calls to all the sub-districts, asking for completed AEFI forms. Supervisors were also encouraged to make frontline HCWs understand that AEFI monitoring and reporting is an integral part of their work and that the reports they generate would be used to improve the immunisation surveillance and not to assess their job performance.

"...we need to be reporting so we will also have to attach some kind of seriousness to it and be on our officers and then make sure every month, they submit reports even if there is none; they should give us zero report. So that will make them aware that they have that responsibility towards us. Also, they should not only report the very severe ones because even if it's a rise in temperature or whether it is just vomiting or whatever; even the slightest event that they could even manage, they should report it to the district level." **(DHMT-Volta Region)**

**Simplifying the AEFI reporting system.** Most respondents thought that simplifying the EPI AEFI reporting forms, making them available to HCWs, providing continuous refresher trainings on how to fill those forms, and creating avenues for feedback at all levels, particularly from the FDA, would improve AEFI reporting.

"We should make the form to be highly and readily available and the filling should be simpler. The form should be simple; we are talking about salient points and avoid the things the person even reporting will not understand. They use words, words that the person even does not understand. Since it is just a report let's just make it simple—one page and they are ready to report. And then as I said, constant reminder and feedback and the person also should get feedback." **(DHMT-Northern Region)**

One HCW suggested possibly using mobile phones or some computer software to report AEFIs.

"Now we have apps, Ghana Health Service should have a platform where you can easily log in and if there is any report it should be referred to the sub-district and steps will be taken; that is the only way to improve reporting. Now with technology, paperwork is no more part of the system because the mother might call you at night and say my child is sick, what should I do? You need to give her prompt attention....I don't think it's an additional cost you just download the app and it is on your phone." **(HCW-Greater Accra)**

**Incentives for frontline HCWs.** Across the regions, HCWs requested incentives to encourage them to report AEFIs. Specifically, they called for health authorities to provide monetary incentives and recognise staff who work hard. The provision of equipment, logistics, and supplies (e.g. motorbikes for follow-ups to clients after vaccination, reporting forms for documenting AEFIs, communication devices for contacting both caregivers and regulatory authorities) were repeatedly mentioned as an effective way of motivating HCWs to monitor and report AEFIs. Frontline HCWs wanted to be encouraged to report AEFIs instead of being vilified by their superiors. The health managers expressed the need for boosting frontline HCW's confidence and allaying their fears of any blame after reporting.

"We just have to allay the fears of health providers that whenever they immunise children, out of ten, one will react to the drug. Adverse events following immunisation are no fault of theirs.

*Because if the child is running temperature after Immunisation is considered an adverse event following immunisation, you have to report. If we build their confidence. . . then they will also be willing to come forward but if we make the thing look like it's a fault-finding thing then why should I report but if we build their confidence level in that aspect and they have the capacity to be reporting, I'm sure they are getting the cases but the fear is that if they report plenty they will say am at fault so they won't report at all"* **(DHMT-Northern Region)**

Putting posters up at the facilities to serve as reminders of AEFI reporting and having focal persons at the facility level to handle AEFI-related issues were among other suggestions to motivate frontline HCWs to report. Importance of the feedback from authorities, receiving the AEFI reports was emphasized repeatedly because this makes HCWs know that the data are being used and serves as a motivation for regular reporting.

**Funding for AEFI reporting.**   Currently, there are no funds within district health budgets to support AEFI reporting. In view of this a member of the RHMT had this to say:

"*If really we want people to report AEFIs, then funds should be set aside. Telling the region this is your funds for AEFIs, if even you will not give it to the region at a go, inform the regions that there are funds set aside for AEFI reporting. Anybody who is in a district that reports we just give it to you to do your follow up. I think it will help to improve the reporting*"
**(RHMT-Greater Accra)**

**Standardising the reporting procedures.**   Respondents wanted the health authorities to take interest in AEFI trends and use legislation to compel HCWs and health managers to present such information as part of their quarterly, mid-year, and annual reports. They also advocated for the standardisation of the medium for reporting AEFIs, across the two EPI and FDA AEFI reporting systems. HCWs stated that currently, paper-based reporting is the most common medium for reporting AEFIs. Standardising the forms and using the same forms across the regions was a common suggestion. The HCWs wanted the forms to be in a tear-out booklet with carbon paper in between the pages for easy duplication. Respondents stated that having a steady supply of reporting forms available at appropriate places would be helpful. They suggested that the AEFI reporting forms could be printed and distributed by the FDA to save the health facilities the cost of photocopying. Facilities that use an online system for reporting AEFIs requested continuous access to the internet.

Phones were repeatedly mentioned as the most convenient method for communicating AEFIs to the next level because it provides the opportunity to receive prompt feedback from the regulators. Reporting of AEFIs and following-up with clients through phone calls was also encouraged. Additionally, some suggested that keeping a list of phone numbers of all their clients should be the policy for HCWs.

"*. . .if we also have electronic way of reporting it, be it through the mobile phone. Because if it's through the mobile phone you can just access the form there and you fill in and then send it. That one can also be faster and then it will aid in reporting." (HCW-Upper East)*

**Review reportable AEFIs to reduce the workload.**   A request was made to review the type of AEFIs that should be reported because the respondents felt that reporting every suspected AEFI per the current guidelines would increase the workload of the HCWs and create indifference to reporting. Specifically, respondents found it unnecessary to report AEFIs that are

already known to be associated with a particular vaccine. HCWs found it awkward to tell care-givers the side effects that the child is likely to suffer upon receiving a particular vaccine, then requesting them to report when it happens.

> "*We had a lengthy discussion; we asked questions especially on this issue of fever because on the AEFI form, we have fever there and so for us, fever is normal so long as especially PENTA is concerned. And so would you have to be filling AEFI forms for every child that you give PENTA to? It was a concern, because we all know fever presents after PENTA and so every child that receives PENTA, we are going to fill AEFI form for that child.* (**HCW-Volta Region**)

## Discussion

Reporting of AEFIs contributes to improved vaccine safety surveillance system and prompt case management [19, 20]. From the perspectives of a broad range of key informants at all lev-els of the vaccine safety system, we found multiple factors (both structural and behavioural), that may impact HCW reporting of AEFI in Ghana. Strategies to improve AEFI reporting sug-gested by participants included both long term (e.g., integration of AEFIs into routine moni-toring and reporting, effective supervision, standardization of reporting procedures across regions and reconsidering what constitutes a reportable AEFI) and shorter term activities (e.g. simplification of reporting procedures, incentives, and capacity building). Implementation of some or all of these strategies could contribute to substantial improvements in their current vaccine safety surveillance system.

Our findings suggest that frontline HCWs generally have limited knowledge of which AEFIs to report. Twene and Yawson reported gaps in the knowledge of HCWs with regards to the definition of AEFI, AEFI investigation and conditions that should be reported as AEFIs in some parts of Ghana [21]. Similarly, in Nigeria, Omoleke et al. reported varied and suboptimal knowledge levels of healthcare providers on AEFI definitions and classifications [22]. Also, our study showed that many frontline HCWs are not familiar with the AEFI reporting proce-dures–what to report, who to report to, and how to report. This is mostly because those in supervisory roles, such as health facility in-charges usually attend the training programs on immunisations and AEFI reporting instead of the frontline HCWs. Consequently, frontline HCWs lack knowledge, which impacts case detection, management, and reporting of AEFIs in general [23, 24]. Decentralising the capacity-building programs on AEFI reporting to trainings at the community level has the potential to increase the participation of more frontline HCWs; hence, the possibility of increased AEFI reporting.

We also found that HCWs have different perceptions about what constitutes a reportable AEFI, which contributes to under-reporting [25]. Some studies have shown that HCWs believe that only proven adverse events should be reported [26]. Unsurprisingly, some HCWs in our study requested a review of reportable events to allow for the reporting of only severe and unexpected adverse events. Additionally, AEFI reporting guidelines appear to conflict with what HCWs are accustomed to during their professionalization process. For instance, HCWs have to be almost certain of a diagnosis before prescribing a drug or referring a patient to a specialist. However, AEFI reporting requires that they report even if they suspect the event is not related to the vaccine [27]. Re-orienting HCWs to think differently about AEFI reporting will require focused training with an understanding of adult learning techniques and clear case definitions.

As some studies have shown, many HCWs who are familiar with the reporting system choose not to report when they encounter AEFIs for fear of consequences and potential legal

liability, and accusations for professional negligence [28–31]. Some HCWs consider reporting an AEFI is "putting your own neck out" for blame by parents and other HCW [25]. Thus, creating a "no blame" culture within the health system for AEFIs is critical. Findings from our study suggest that some HCWs refuse to report AEFIs because they believe AEFI reports are used for professional evaluations rather than improving vaccine surveillance and safety.

Reporting requirements of AEFI by national FDAs can be complex and cumbersome. Under-reporting is common when reporting tools and processes are unclear [32], especially if these reports are not perceived as part of HCW's core responsibilities [26, 30]. In Ghana, most frontline HCWs are community health officers (CHOs) who have the primary responsibility of implementing Ghana's flagship program for primary health care—the CHPS program. Within this program, CHOs deliver basic health services, including immunisations to remote community members [33]. Research in other settings suggests that frontline HCWs who are overburdened due to patient load and administrative duties tend to either forget to report AEFIs or treat them as "the other thing" and prioritise other tasks within the facility [26, 30, 34]. Currently, routine reports submitted by CHOs do not include AEFIs. However, when AEFIs occur, primary health care workers are expected to diagnose, provide first hand information, counsel the patient and initiate appropriate management [35]. Therefore, suggestions by respondents to integrate AEFI reporting into the routine facility and sub-district reports are not out of place. The participation of primary health care nurses in reporting AEFIs will also improve patient safety and reduce cost associated with AEFIs [36].

At the primary health care level, tools available for reporting AEFIs are largely paper-based. Electronic data capture is currently being used at the district level where information officers key in aggregate data from community health facilities. Also, officials of the FDA in urban areas such as Accra and Kumasi have access to the WHO Vigi-base to regularly input AEFIs. AEFI reporting would however benefit from expanding the use of electronic systems to facilitate AEFI reporting by caregivers and HCWs at the community level. This could be integrated into an active surveillance system that uses Short Message Service (SMS) to potentially permit more rapid identification of emerging safety signals as shown in places like Cameroun [37]. Concerns about the use of electronic systems have focused on cost. However, in Cameroun, a pilot test of telephone "beep" significantly increased AEFI reporting within communities at an affordable cost [37].

The absence of feedback on AEFI reports from higher authorities is an important factor that discourages HCWs from reporting AEFIs. Studies have demonstrated that HCWs are less likely to report AEFIs due to lack of feedback [31, 38]. In some cases, the only feedback HCWs receive is blame and accusations for reporting an AEFI regardless of the cause [25]. Feedback can help identify what works in health care services and the quality of care provided to patients. When serious and/or unexpected adverse events occur, HCWs need to obtain more detailed medical information to assess causality which makes feedback from both the EPI and FDA a necessity. This would raise awareness and knowledge for vaccine-associated risks among HCWs. In fact, this is consistent with the Ghana FDA regulations, which states that the FDA is to provide feedback for the AEFIs to the reporter within one month of evaluation by the authority [17]. Based on our findings, adhering to this guideline appears to be a challenge for the Ghana FDA, and the lack of feedback may contribute to underreporting, which is consistent with other research on AEFI reporting in Africa [31].

Lack of motivation among frontline HCWs has contributed negatively to AEFI surveillance, especially on reporting of adverse events in some African countries [29]. The efficiency of public health services in Ghana has been linked to motivational packages, such as financial incentives, transportation, and accommodation [39]. Respondents in our study reported there is no motivation to report AEFIs. The absence of monitoring and supervision, and costs incurred by

facilities for reporting AEFIs affect motivation. Even participation in training workshops, which could serve as motivation is not afforded to frontline HCWs who are the primary reporters of AEFIs. New vaccine introductions could serve as avenues for additional training for frontline HCWs on AEFI. HCWs know they are supposed to report, but they feel discouraged to report AEFIs because there is generally no appreciation for their effort [38]. Putting in place motivational schemes (e.g. such as positive feedback, certifications, etc.) that recognize the efforts of the facilities and individuals who report AEFIs could increase reporting.

## Study strengths and limitations

The study sample consists of frontline HCWs, health managers, and regulators from four out of the ten regions in Ghana. This exploratory study reflects the perspectives of these individuals on the AEFI reporting process.

One of the limitations of the study is that the research approach did not allow for confirmation of information from respondents, such as the availability of reporting forms, provision of feedback by the regulators, and the attendance sheets of trainees at recent trainings on AEFIs. Thus, the study is limited by the self-reporting of respondents, which increases the possibility of reporting bias. Additionally, data being collected by numerous research assistants may have distorted the presentation of the questions to the respondents. However, we believe that this was minimized by the fact that data collectors were graduate-level research assistants with experience in qualitative research, and they have received a standard training that included role plays and pre-tests of the study guides.

## Conclusions

Immunisation is an effective way of saving lives; however, poor monitoring and management of AEFIs can affect public trust in immunisation, especially for new vaccines. Reporting of AEFI by HCWs is crucial for prompt response and management of AEFI and for monitoring the vaccine safety post-licensure. The challenges frontline HCWs face can be addressed by building their capacity in reporting, re-orienting them on reportable AEFIs, providing motivation and feedback, and simplifying the reporting process. The widespread use of mobile phones presents an opportunity for Ghana FDA and EPI. The feasibility of using electronic reporting systems should be explored to increase AEFI reporting in Ghana.

## Supporting information

**S1 Data.**
(ZIP)

## Acknowledgments

We would like to thank in-country stakeholders, including EPI, the Ghana Food and Drugs Authority, Navrongo Health Research Centre, and the African Collaborating Centre for Pharmacovigilance. We also would like to thank the data collectors who worked tirelessly under difficult field conditions during data collection. Additionally, we greatly appreciate Mohamed Jalloh, Abigail Shefer, David Fitter, and Sureyya Hornston's critical review of this qualitative manuscript.

## Author Contributions

**Conceptualization:** Raymond Akawire Aborigo, Jane F. Gidudu.

**Data curation:** Raymond Akawire Aborigo, Paul Welaga, Abraham Oduro, Joseph Opare, Hilda Ampadu.

**Formal analysis:** Raymond Akawire Aborigo.

**Funding acquisition:** Jane F. Gidudu.

**Investigation:** Raymond Akawire Aborigo.

**Methodology:** Raymond Akawire Aborigo, Paul Welaga, Anna Shaum, Joseph Opare, Alex Dodoo, Hilda Ampadu, Jane F. Gidudu.

**Project administration:** Raymond Akawire Aborigo.

**Supervision:** Raymond Akawire Aborigo, Abraham Oduro.

**Validation:** Raymond Akawire Aborigo, Paul Welaga, Abraham Oduro, Anna Shaum, Joseph Opare, Alex Dodoo, Hilda Ampadu, Jane F. Gidudu.

**Writing – original draft:** Raymond Akawire Aborigo.

**Writing – review & editing:** Paul Welaga, Abraham Oduro, Anna Shaum, Joseph Opare, Alex Dodoo, Jane F. Gidudu.

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
