## [Decision Letter · Decision Letter 0]

1 Aug 2022

PONE-D-21-25213Optimising reporting of adverse events following immunisation by healthcare workers in Ghana: qualitative study in four regionsPLOS ONE

Dear Dr. Aborigo,

Thank you for submitting your manuscript to PLOS ONE. After careful consideration, we feel that it has merit but does not fully meet PLOS ONE’s publication criteria as it currently stands. Therefore, we invite you to submit a revised version of the manuscript that addresses the points raised during the review process.

Look  that there are some revision to do, but after done the manuscript may be accepted.

We look forward to receiving your revised manuscript.

Kind regards,

Ricardo Q. Gurgel, PhD

Academic Editor

PLOS ONE

Journal Requirements:

3. Thank you for stating the following in the Acknowledgments Section of your manuscript: "We would like to thank in-country stakeholders, including EPI, the Ghana Food and Drugs Authority, Navrongo Health Research Centre, and the African Collaborating Centre for Pharmacovigilance. We also would like to thank the data collectors who worked tirelessly under difficult field conditions during data collection. Additionally, we greatly appreciate Mohamed Jalloh, Abigail Shefer, David Fitter, and Sureyya Hornston’s critical review of this qualitative manuscript. 

The research was supported by the Centers for Disease Control and Prevention/the Agency for Toxic Substances and Disease, Atlanta, USA."

Please remove any funding-related text from the manuscript and let us know how you would like to update your Funding Statement. Currently, your Funding Statement reads as follows: "No -The funders had no role in study design, data collection and analysis, decision to publish, or preparation of the manuscript."

5. Please upload a new copy of Figure 1 as the detail is not clear. Please follow the link for more information: https://blogs.plos.org/plos/2019/06/looking-good-tips-for-creating-your-plos-figures-graphics/" https://blogs.plos.org/plos/2019/06/looking-good-tips-for-creating-your-plos-figures-graphics/

6. Please include your tables as part of your main manuscript and remove the individual files. Please note that supplementary tables (should remain/ be uploaded) as separate "supporting information" files"

Reviewers' comments:

Reviewer's Responses to Questions

**Comments to the Author**

1. Is the manuscript technically sound, and do the data support the conclusions?

Reviewer #1: Yes

Reviewer #2: Yes

2. Has the statistical analysis been performed appropriately and rigorously? 

Reviewer #1: Yes

Reviewer #2: N/A

3. Have the authors made all data underlying the findings in their manuscript fully available?

Reviewer #1: Yes

Reviewer #2: No

4. Is the manuscript presented in an intelligible fashion and written in standard English?

Reviewer #1: Yes

Reviewer #2: Yes

5. Review Comments to the Author

Reviewer #1: Dear authors. The research carried out with a broad character on vaccine notification is valid, relevant, necessary and very interesting. It would only reduce the number of keywords. The introduction is well written, relating to the topic. It's good, comprehensive, and complete. The materials and methods, the description is a little confusing, although it has a table that makes it a little easier. The part that refers to the data collection technique needs to be better clarified. the data analysis refers to the use of NVivo software for qualitative research and the presence of two judges. Discussions in a separate session of the results were made a synthesis. very interesting needed in qualitative work, although here somewhat repetitive

Please be advised that out of consideration for the authors, if we do not receive your review or a response within the next three days, we must assume that you are not available to complete this assignment and will remove it from your account. The Academic Editor will then proceed to render a decision based on previously secured reviews and their own expertise. If it is your intention to submit your review, please do let us know.

Reviewer #2: Reviewer's comments

1. Line 64: kindly amend the sentence.

2. Line 103: Reference should be properly placed.

3. References needs to be correctly inserted. For example, references 10-13.

4. How were the districts chosen?

5. Line 140 should be proofread and edited.

6. Line 141: “According to the FDA guidelines, a suspicion alone is enough grounds for 142 reporting.” This should be referenced

7. Generally, four levels 145 of reporting are used during mass immunization campaigns. These are the health 146 worker, the district, the regional and the national levels. These need to be adequately broken down as it is confusing in its present state. Can an AEFI report from the region? Quite unlikely!

8. Line 154: spelling of vaccinee is wrong. Kindly correct it

9. Inconsistent capitalization of proper noun

10. Is the Regional Focal Person the same the Regional EPI Coordinator? There should be consistency in the use of these designations.

11. The manuscript should be properly formatted. For example, see page 12.

12. “Data 240 collectors conducted 116 KIIs from December 2017 to January 2018 with respondents 241 at their work-places in four regions.” Why these large number of respondents for a qualitative survey keeping in mind the concept of saturation?

13. Discussion: comparison was not made with studies conducted within the subregion or elsewhere regarding the understanding or knowledge of AEFI by the health workers and immunization manager. You may check out this study: Omoleke et al. BMC Health Services Research (2022) 22:741 https://doi.org/10.1186/s12913-022-08133-9

14. “Some studies have shown that HCWs believe that only proven adverse events should be reported”. What does the author mean by proven adverse events?

15. “Re-orienting HCWs to think differently about AEFI 581 reporting will require focused training with an understanding of adult learning techniques and clear case definitions.” This assertion could also be substantiated by findings from the recently published study from Nigeria that found suboptimal understanding of AEFI about health workers and called for a regular refresher training on the subject.

16. Again, the referencing for the entire manuscript needs to be reworked. Ogunyemi et al was cited twice- reference number 31 and 33.

17. “The participation of primary health care nurses 608 in reporting AEFIs will also improve patient safety and reduce cost associated with AEFIs[37]” Why singling out community health nurses when the authors had clearly mentioned elsewhere that CHO are mainly responsible for implementing Ghana PHC flagship programme at the health facility level?

18. Line 648: New vaccine introductions could serve as avenues for additional training for frontline HCWs on AEFI. This was not completed- how does it affect motivation of the HCW to report?

19. Line 670 “Immunisation is an effective way of saving lives”- this should be re-written.

6. PLOS authors have the option to publish the peer review history of their article (what does this mean?). If published, this will include your full peer review and any attached files.

Reviewer #1: **Yes: **Maria Viviane Lisboa de Vasconcelos

Reviewer #2: **Yes: **Dr Semeeh Akinwale OMOLEKE

---

## [Author Response · Author response to Decision Letter 0]

20 Oct 2022

Reviewers' comments:

Reviewer's Responses to Questions

5. Review Comments to the Author

Reviewer #1: Dear authors. The research carried out with a broad character on vaccine notification is valid, relevant, necessary and very interesting. It would only reduce the number of keywords. The introduction is well written, relating to the topic. It's good, comprehensive, and complete. The materials and methods, the description is a little confusing, although it has a table that makes it a little easier. The part that refers to the data collection technique needs to be better clarified. the data analysis refers to the use of NVivo software for qualitative research and the presence of two judges. Discussions in a separate session of the results were made a synthesis. very interesting needed in qualitative work, although here somewhat repetitive

Please be advised that out of consideration for the authors, if we do not receive your review or a response within the next three days, we must assume that you are not available to complete this assignment and will remove it from your account. The Academic Editor will then proceed to render a decision based on previously secured reviews and their own expertise. If it is your intention to submit your review, please do let us know.

Reviewer #2: Reviewer's comments

1. Line 64: kindly amend the sentence.

Response: We thank the reviewer for catching the typo. The sentence has been edited appropriately. 

2. Line 103: Reference should be properly placed.

Response: The reference has been placed appropriately.

3. References needs to be correctly inserted. For example, references 10-13.

Response : This has been done

4. How were the districts chosen?

Response: Under the section on sampling we stated that “In each region, we randomly selected five districts and conducted KIIs with two members of the DHMT (the public health nurse and the health information officer) and the EPI coordinator who, in most cases, was the disease control officer”.

5. Line 140 should be proofread and edited.

Response: The sentence has been edited appropriately. We thank the reviewer for pointing this out.

6. Line 141: “According to the FDA guidelines, a suspicion alone is enough grounds for 142 reporting.” This should be referenced

Response: The reference has been inserted

7. Generally, four levels 145 of reporting are used during mass immunization campaigns. These are the health 146 worker, the district, the regional and the national levels. These need to be adequately broken down as it is confusing in its present state. Can an AEFI report from the region? Quite unlikely!

Response: The sentence has been edited appropriately. We intended capturing the flow of information on AEFIs related to mass immunization campaigns. 

8. Line 154: spelling of vaccinee is wrong. Kindly correct it

Response: This has been corrected.

9. Inconsistent capitalization of proper noun

Response: This has been corrected. Thanks for catching that.

10. Is the Regional Focal Person the same the Regional EPI Coordinator? There should be consistency in the use of these designations.

Response: We have the Regional EPI Coordinator and the FDA Regional Focal Person. These are two different designations. There is no such position as Regional Focal Person.

11. The manuscript should be properly formatted. For example, see page 12.

Response: This has been done

12. “Data 240 collectors conducted 116 KIIs from December 2017 to January 2018 with respondents 241 at their work-places in four regions.” Why these large number of respondents for a qualitative survey keeping in mind the concept of saturation?

Response: We do recognize that some of the numbers were too large and could have been reduced without compromising the quality of our data or attaining thematic saturation. However, you will realize that we had different categories of respondents who were spread across a wide range of locations and we were interested in triangulating across data sources and to explore geographical variations that may be related to AEFI reporting. These factors contributed to the large numbers.

13. Discussion: comparison was not made with studies conducted within the subregion or elsewhere regarding the understanding or knowledge of AEFI by the health workers and immunization manager. You may check out this study: Omoleke et al. BMC Health Services Research (2022) 22:741 https://doi.org/10.1186/s12913-022-08133-9

Response: Thank you for the reference. We have referenced the paper in our manuscript.

14. “Some studies have shown that HCWs believe that only proven adverse events should be reported”. What does the author mean by proven adverse events?

Response: Adverse events that HCWs know to be related to the vaccine. The word proven has been replaced appropriately.

15. “Re-orienting HCWs to think differently about AEFI 581 reporting will require focused training with an understanding of adult learning techniques and clear case definitions.” This assertion could also be substantiated by findings from the recently published study from Nigeria that found suboptimal understanding of AEFI about health workers and called for a regular refresher training on the subject.

Response: This reference has been added.

16. Again, the referencing for the entire manuscript needs to be reworked. Ogunyemi et al was cited twice- reference number 31 and 33.

Response: The references have been reworked as suggested.

17. “The participation of primary health care nurses 608 in reporting AEFIs will also improve patient safety and reduce cost associated with AEFIs[37]” Why singling out community health nurses when the authors had clearly mentioned elsewhere that CHO are mainly responsible for implementing Ghana PHC flagship programme at the health facility level?

Response: They are the frontline providers for immunizations but they are currently missing in the reporting system. Also, they are the closest to the clients and therefore accessible when clients want to report AEFIs. They are therefore better positioned to receive and report AEFIs.

18. Line 648: New vaccine introductions could serve as avenues for additional training for frontline HCWs on AEFI. This was not completed- how does it affect motivation of the HCW to report?

Response: The preceding sentence which reads “Even participation in training workshops, which could serve as motivation is not afforded to frontline HCWs who are the primary reporters of AEFIs” explains how training opportunities motivate HCWs.

19. Line 670 “Immunisation is an effective way of saving lives”- this should be re-written.

Response: This has been revised appropriately.

We thank the reviewers for their useful feedback, and we are happy to address any further queries that they may have.

---

## [Editor Report · Decision Letter 1]

24 Oct 2022

Optimising reporting of adverse events following immunisation by healthcare workers in Ghana: A qualitative study in four regions

PONE-D-21-25213R1

Dear Dr. Aborigo,

We’re pleased to inform you that your manuscript has been judged scientifically suitable for publication and will be formally accepted for publication once it meets all outstanding technical requirements.

Kind regards,

Ricardo Q. Gurgel, PhD

Academic Editor

PLOS ONE
---

## [Editor Report · Acceptance letter]

26 Oct 2022

PONE-D-21-25213R1 

Optimising reporting of adverse events following immunisation by healthcare workers in Ghana: A qualitative study in four regions 

Dear Dr. Aborigo:

I'm pleased to inform you that your manuscript has been deemed suitable for publication in PLOS ONE. Congratulations! Your manuscript is now with our production department. 

Kind regards, 

on behalf of

Professor Ricardo Q. Gurgel 

Academic Editor

PLOS ONE